# Reforms in China’s Vaccine Administration—From the Perspective of New Governance Approach

**DOI:** 10.3390/ijerph20043450

**Published:** 2023-02-16

**Authors:** Lin Tang, Lingling Zhang

**Affiliations:** KoGuan School of Law, China Institute for Smart Court, Shanghai Jiao Tong University, Shanghai 200030, China

**Keywords:** decentralization and privatization, healthcare, new governance, lot release, vaccine circulation

## Abstract

Recent vaccine scandals have overshadowed China’s accomplishments in public health, triggering discussions on the causes of vaccine incidents. This study aims to review the development of China’s vaccine administration, find out the causes of recurring vaccine incidents in the past decades, and propose a new governance approach to vaccine administration in the context of a public resource trading system. We collect and analyze relevant legal frameworks and data from legislative materials, government documents, press releases, and reports from the World Health Organization. In essence, it is the combination of the lagging legal system and the absence of information technology infrastructure in the process of vaccine administration reform that has led to the recurrence of vaccine incidents. Though the vaccine incidents occurred concentratedly in phases of production and lot release, and circulation, it is necessary to examine the whole life cycle of vaccine administration. The enactment of the Vaccine Administration Law outlines a supervision framework, which utilizes the Whole Process Electronic Traceability System and Whole Life-cycle Quality Management System to achieve the interconnection of all aspects of vaccine administration. The reform of China’s vaccine administration is essentially a balance between efficiency and safety, which also represents the interaction between marketization and administrative supervision.

## 1. Introduction

The past decades have witnessed a myriad of vaccine incidents across China, significantly aggravating public distrust in vaccination programs and causing decreased use of vaccines [1]. In 2016, the vaccine scandal in Shandong Province, fueled by exaggerated reporting from various media outlets, aroused widespread public panic in the country [2]. The most recent vaccine safety incident, in Changsheng in 2018, further propelled the vaccine administration into the national spotlight, as the atmosphere of vaccine hesitancy had diffused prevalently and was proving to be a threat to China’s public health accomplishments [3].

On the one hand, most previous studies of the vaccine administration in China proposed that the profit-seeking nature of enterprises, the inadequacy and deficits of governmental supervision, and the rent-seeking of local governmental officials are the root causes of the vaccine incidents [3,4,5]. However, the existing accounts failed to draw on any systematic research into the evolution of China’s vaccine administration, neglecting the fact that vaccine administration involves the supervision of the entire life cycle of the vaccine.

On the other hand, China has adopted the traditional command-and-control mode in vaccine administration for decades, especially in the aspects of vaccine circulation. Scholars have long criticized the traditional command-and-control regulation for its shortcomings and frequent failures. The pervasive inadequacies of market self-ordering are also well-documented. Regulators worldwide are figuring out novel third-way approaches to regulation, collectively referred to as the new governance model, in response to the ongoing need for government intervention, the limitations of traditional command-and-control regulation, and the mounting pressure to liberalize markets [6]. Policymakers and academics are concentrating their attention on new governance as a collection of legal strategies outside the command-and-control mode, aimed at improving the efficacy and legitimacy of social regulation, both in theory and in practice [7].

In retrospect, on China’s vaccine administration in the last two decades (as shown in Table 1), it is not hard to see that vaccine incidents occurred concentratedly in production and lot release, and circulation. As the world’s largest vaccine producer, with 45 vaccine manufacturers, China kept the amount of vaccine lot release at 0.5–1 billion bottles per year from 2010 to 2019 [8]. Facing such an enormous amount of vaccine lot release, a well-designed system of vaccine quality management and distribution is the key to the vaccine administration, which requires the legislation to advance with time, and the construction of the required infrastructure.

Therefore, concerning the debate on vaccine incidents in China, the first and foremost question is how to correctly analyze China’s efforts in vaccine administration over the past 30 years. To put it another way, it is necessary to delve into multiple aspects of the vaccine administration, and disentangle the reality of China’s vaccine administration from the history entwined with vaccine scandals. This paper contributes to the debate by scrutinizing the development of lot release and the mode of China’s vaccine circulation with new governance approaches. The next section reinterprets selected vaccine incidents in China, and exposes the vulnerabilities of the vaccine administration at that time, to lay the foundation for the analysis of the corresponding legislative progress, as well as the development of platform infrastructures. After proposing the structurally integrated decentralization model, based on the practice of the Public Resource Trading system, this paper then presents a systemic view on China’s Vaccine Administration Law, and ends with concluding remarks on the prospects of China’s vaccine administration.

## 2. The Legal Interpretation of Vaccine Incidents and Revelation of Potential Conflicts

### 2.1. The Crisis Simmered in Planned Mode of Vaccine Circulation: Vaccine Incident in Suqian City and Healthcare Reform

In the 1980s, China rapidly achieved universal childhood immunization after launching the Expanded Program on Immunization, which meant that the vaccine coverage (including BCG, polio, diphtheria, pertussis and tetanus, and measles) for 12-month children reached 85% across all provinces and counties (Figure 1). In October 2000, the National Polio Eradication Committee of China, and the World Health Organization (WHO) Western Pacific Region, jointly confirmed that the spread of the wild poliovirus in China had been eradicated and China had become a polio-free country [9]. Besides, since the hepatitis B vaccine was included in the National Immunization Program in 2002, the coverage of hepatitis B immunization among 12-month children in China had reached 91% by 2006 (Figure 1), and 11.1 million children living in destitute areas had also received the hepatitis B vaccination [10].

The enactment of the Law on Prevention and Treatment of Infectious Diseases in 1989, and Measures for Implementation of Law on Prevention and Treatment of Infectious Diseases in 1991, formally established the planned prophylactic vaccination system, under which bacterin, vaccines, and other biotic products should be ordered by the Anti-epidemic Stations of all provinces, autonomous regions, and municipalities directly under the Central Government from the biotic product production units [11]. Moreover, all Anti-epidemic Stations are responsible for the ordering, distribution, and storing level by level, and all kinds of healthcare facilities, anti-epidemic institutions, and their personnel must use the preventive biotic products that are distributed by Anti-epidemic Stations level by level. Nevertheless, all Anti-epidemic stations at each level would charge a markup on the sales of vaccines, to compensate for low-price medical services. The low medical service fee scheme stemmed from the planned economy era in the 1950s, setting the service price lower than the cost to guarantee universal access to healthcare.

Ostensibly, before the first reported vaccine incident in 2004, China enjoyed a great reputation for improving public healthcare. However, an enforced action carried out by the Food and Drug Administration (hereinafter, FDA) of Suqian city (a prefectural city in northern Jiangsu Province) confiscated a large number of non-qualifying vaccines in 2004, arousing widespread public concern, since the majority of those vaccines had already been inoculated into newborn infants [12]. From 2003 to 2004, the Maternal and Child Care Service Center in Suqian purchased approximately 6000 vaccines in a cash settlement with the salesman who provided an invoice from the Chuzhou Pharmaceutical Technology Development Corporation rather than from the official channels of the Anti-epidemic Station. This corporation was just run as a drug wholesaler [12]. In addition, those non-qualifying vaccines were not kept in professional cold chain facilities during transportation, which led to degraded or even ineffective vaccines [12].

However, there is a critical question regarding the circulation of vaccines: why did the Maternal and Child Care Service Center risk purchasing defective vaccines from an unreliable wholesaler instead of the Anti-epidemic Station, which could at least guarantee the safety of vaccines? There are two main causes: the drastically retrenched health budget caused by the healthcare reform in the 1980s, and the end of the vaccine monopoly in the 1990s.

#### 2.1.1. Load Shedding: The Radical Marketization in Healthcare Reform

In tandem with the economic reforms of the early 1980s, China initiated the first round of healthcare reform in 1985, which mainly aimed at establishing various healthcare institutions through multiple channels. Worse still, the fiscal and taxation reform in 1994, i.e., Tax-Sharing Reform, that readjusted the tax distribution and structure between the Central and local governments, plunged the government health operational expense, as a percentage of total fiscal spending, into recession, from 2.68% in 1981 to 1.59% in 2002 [13]. From 1978 to 1999, the share of total health expenditure from the Central government plummeted from 32% to 15%. In contrast, the out-of-pocket spending as a share of total healthcare expenditure indicated that the health burden was transferred to provinces and individuals, ballooning from 20% in 1978 to 58% in 2001 [13].

Against the market-oriented reform and unprecedented fiscal burden, a radical healthcare movement was also unveiled in Suqian city in 2000. Suqian adopted an over-the-top development model, of which healthcare reform formed a key part of the urban development policy to catch up with economically developed areas [14]. As of 2004, medical service institutions at all levels in Suqian city, including 124 township health service centers and 10 hospitals above the county level, had been purchased or operated by private capital, symbolizing the complete withdrawal of government capital from healthcare undertakings [14]. The government substantially cut the fiscal budget for public-welfare non-profit institutions, including institutions for maternity and child care, health emergency response, and blood collection and supply [14]. By way of illustration, the payroll of the Maternal and Child Care Service Center in Suqian accounted for almost 50% of the total government subsidy in 2004, which, consequently, started to reduce the costs of drug procurement with recourse to drug wholesalers [12].

#### 2.1.2. The End of State’s Monopoly of Non-Immunity-Planning Vaccines

In 1989, the Ministry of Health integrated the Central Epidemic Prevention Department and six local research institutes (Beijing, Shanghai, Wuhan, Chengdu, Changchun, and Lanzhou Institute of Biological Products) to form the China Biological Products General Corporation (CBPGC), the predecessor of China National Biotec Corporation (CNBC), which almost monopolized the supply of national vaccines at that time [15]. Accompanied by the full implementation of the Expanded Program of Immunization (EPI) in the 1990s, the administrative control over access to the vaccine industry began to relax, which was effectively a license for private and even foreign capitals to step into the vaccine field, and struck at the very foundation of the monopoly of non-immunity-planning vaccines. With the advance of the vaccine market, and rising demands for vaccination, both vaccine production and wholesale enterprises were allowed to sell non-immunity-planning vaccines to disease prevention and control institutions and inoculation entities in 2005, formally declaring the end of the monopoly of non-immunity-planning vaccines [16].

### 2.2. Vaccine Incidents Alternately Occurred in the Process of Production and Circulation

#### 2.2.1. The Faulty Vaccine Incident of Yanshen Biological Technology Stock Company in 2010

It was the State Food and Drug Administration (hereinafter, SFDA) that declared there were severe quality problems with four batches (179,952,000 copies in total) of rabies vaccine for human use, produced by Yanshen Biological Technology Stock Company (hereinafter, Yanshen) between July and October in 2008, after an unannounced inspection of vaccines. The exposure of the faulty vaccines incident occurred almost one year after the faulty vaccines were sold into several provinces, including Fujian, Hebei, Anhui, and Jiangsu [17].

Since the lot release regime for biological products was established in 2004, all types of vaccines must go through the sample inspection conducted by the National Institutes for Food and Drug Control (hereinafter, NIFDC), when each product lot is marketed or even imported [18]. Generally, lot release involves four steps: application, examination, inspection, and release. In particular, the sampling procedure during the application, which used to be conducted by accredited supervisors, is a decisive factor in ensuring the safety and effectiveness of vaccine products, in that the authenticity of samples directly dictates the reliability of the inspection results [18]. But, rashly, due to the lack of comprehensive regulations on sampling procedures, the SFDA delegated the responsibility of sampling to accredited supervisors who were also in charge of on-the-spot supervision and inspection of the pharmaceutical manufacturing.

A closer examination of the regime of the accredited supervisors would reveal a significant drawback of the sampling procedure. In principle, accredited supervisors should be assigned to the same drug producer for no more than two years continuously, which means they might not be a professional in vaccine sampling, as they cannot always fulfill their duties in vaccine manufacturers, not to mention the supervision and inspection duties over the four manufacturing behaviors of the enterprise and the work from the SFDA that they are assigned to [18]. In this vaccine incident, after comparing the stock samples in the factory of Yanshen with the samples provided for lot release, the NIFDC concluded that there was an apparent difference in terms of efficacy indicators, which means Yanshen tampered with the process of vaccine sampling [17]. It is affirmative that the institutional deficiencies of sampling and inspection discussed above jointly contributed to this calamitous vaccine incident.

#### 2.2.2. The Shandong Illegal Vaccine Sales Incident in 2016

In contrast to the vaccine incident of Yanshen, the one in Shandong mainly involves the process of vaccine circulation. On 18 March 2016, Shandong police cracked down on an illegal vaccine sales case, in which $88 million worth of vaccines was sold to 24 provinces and cities over five years [19]. Though the ringleaders, a former hospital pharmacist and her daughter, purchased vaccines from qualified domestic manufacturers, they neither adequately kept the vaccines refrigerated nor transported them in approved conditions, which led to reduced vaccine efficacy and even severe allergic reactions [19].

The judgment of the first instance from Jinan Intermediate People’s Court, Shandong Province, reveals that those vaccines, including the rabies vaccine (Vero Cell) for human use, Haemophilus influenzae type B vaccine, rotavirus vaccine for oral use, etc., are Class-II vaccines, with which citizens are voluntarily inoculated at their own expense [19]. Since 2005, both drug producers and wholesalers have been allowed to sell Class-II vaccines to disease prevention and control institutions and inoculation entities. Though the legislative purpose of this regulation was meant to cut the overlong circulation chain of vaccines among hierarchical Anti-epidemic Stations (the predecessor of the Center for Disease Control and Prevention) and break the monopoly of vaccines, third parties without the permit for pharmaceutical business operation took advantage of the relaxed administration of vaccine circulation, exploited the relationship between wholesalers and drug manufacturers, and transported vaccines without qualified cold chain facilities.

Therefore, the occurrence of vaccine incidents in Shandong, in a sense, appears to be a continuum of the one in Suqian city, with the unsettled issue of vaccine circulation. The Regulation on the Administration of Circulation and Vaccination of Vaccines in 2005, in essence, as the official confirmation of market-oriented vaccine administration, improved the supply side of vaccines by retreating the role of the State while introducing market forces. Yet the negligent supervision of vaccine circulation, which had been neglected all this time, led not only to recurring vaccine incidents in the process of circulation, but in escalating harm to public health.

### 2.3. General Outbreak: The Changsheng Vaccine Incident in 2018

Before the National Medical Products Administration (NMPA) launched the unannounced inspections over Changsheng Biotechnology Co., Ltd., Changsheng was once China’s second-largest manufacturer of the rabies vaccine for human use, with CNY 1.55-billion annual revenue in 2017 [20]. On 15 July 2018, the National Medical Products Administration suddenly declared that Changsheng had falsified production data for rabies vaccines and revoked its GMP license. To be more specific, Changsheng blended different batches of rabies vaccines, concentrated the solution, and even used expired ones to produce new batches of rabies vaccines with fabricated production batch numbers [20].

Generally, the inspection results of lot release consist of three categories of indicators: safety indicators (including sterility, pyrogen, bacterial endotoxin, and abnormal toxicity), efficacy indicators (polysaccharide content and potency test), and quality indicators (other appearance properties) [21]. Except for the safety and quality indicators, it has been reported that, regarding efficacy indicators, the NIFDC adopted the internationally accepted practice, i.e., randomly selecting 5–10% of sample batches for efficacy testing on average [22]. Unlike the faulty vaccine incident of Yanshen, Changsheng survived the efficacy testing in 2018 because those substandard rabies vaccines happened to be outside the 5–10% of sampling range [20]. As a listed company on the Shenzhen Stock Exchange, the Changsheng vaccine incident exposed a systematic failure to supervise both the manufacturing process and sample inspection of lot release.

## 3. Integrated System of Vaccine Administration

### 3.1. The Evolution of Vaccine Circulation: From Unified Mode to Public Resources Trading Mode

As the watershed of vaccine circulation, the promulgation of the Regulation on the Administration of Circulation and Vaccination of Vaccines in 2005 represents the transformation of vaccine circulation from a unified model to a decentralized one. In the planned economy era, all types of vaccines were purchased uniformly by the provincial disease control and prevention institutions, and supplied level by level based on the “Province-City-County” model. Thanks to the advance of the healthcare reforms in the 1980s, and the introduction of private and foreign capital into the field of vaccine manufacturing in the 1990s, the State Council made reforms on the supply side by permitting both vaccine manufacturers and qualified wholesalers to sell Class-II vaccines to all-level disease control and prevention institutions and inoculation entities in 2005 [14]. However, owing to the deficiency in proper supervision of vaccine circulation, there have been constant vaccine incidents in the past decades, causing immeasurable physical and mental damage to vaccinees, and thereafter even vaccination hesitancy across the country.

After the vaccine incident in Shandong, the State Council overhauled the decentralized model of vaccine circulation and issued the revised Regulation on the Administration of Circulation and Vaccination of Vaccines (2016 revision), of which Article 10 and Article 15 jointly require provincial disease prevention and control institutions to centrally organize all procurements of Class-II vaccines on provincial public resource trading platforms, while vaccine production enterprises should directly distribute vaccines to county disease prevention and control institutions or authorize qualified enterprises with cold chain storage and transport conditions for vaccines. As to the “last mile” distribution, county-level institutions are responsible for supplying inoculation entities within their respective administrative regions. Consequently, all vaccine wholesalers have been completely kicked out since 2016. In addition, Article 18 rules that all disease prevention and control institutions should establish records on purchase, storage, distribution, and supply, while ensuring consistency among bills, account books, goods, and payments.

Thus far, Article 10, Article 15 and Article 18 together made up the “single invoice” system for vaccine procurement, which means that Class-II vaccines are invoiced only once, when they are listed, and county disease control and prevention institutions purchase vaccines directly from the vaccine manufacturers through provincial public resource trading platforms.

#### 3.1.1. The Architecture of the Public Resource Trading Platform System

Before discussing the circulation mode change from 2016, it is essential to minutely analyze the system of the public resource trading platform, to better understand the mechanism of vaccine circulation under the public resource trading platform. Specifically, the trading platform system, as an embodiment of the concept of unified public resource trading, can be traced back to the public resource trading center set up in Shaoxing city in 2002, which integrated bidding for engineering construction projects, granting of land use rights and mining rights, trading of state-owned property rights, and government procurement into the unified transaction platform [23]. Unlike the traditional top-down healthcare reforms in China, the establishment of a unified public resource trading system follows a unique bottom-up mode, which means it is not solely the State’s plan, but the practices of local governments that promote the development of a unified public resource trading system. This can be illustrated briefly by the preliminary statistics at the provincial, municipal, and county levels, in which there had been about 1203 centralized markets for public resource transactions and the construction of the corresponding electronic trading platform was also being accelerated.

In 2016, the Interim Measures for the Administration of Public Resources Trading Platforms (hereinafter, Interim Measures) was issued by 14 ministries and commissions of the State Council, which elaborately provides rules about the operations, services, supervision, and management of the public resource trading platform, marking the initial establishment of the public resource trading platform system at the state level. From the perspective of architecture, Interim Measures stipulates the public resource electronic trading system (E-trading system), public resource trading (PRT) electronic service system (E-service system), and PRT electronic regulatory system (E-regulatory system). In general, as shown in Figure 2, the E-trading system and E-service system are operated and maintained by the PRT platform operation service institute, i.e., the entity established or designated by governments, which provides public services to related market participants, while the E-regulatory system is in charge of administrative departments. In terms of the types of resources that can be traded, the E-trading system not only covers bidding for construction projects, granting of land use rights and mining rights, trading of state-owned property rights, and government procurement, but also incorporates all other types of trading of public resources based on actual circumstances.

As to the operation mechanism of PRT, the E-service system plays the role of a hub that connects the E-trading system, E-regulatory system, and other external electronic systems of relevant government departments. Regarding information disclosure, the E-trading system should disclose general information [24] and trading information [24] to the public through the E-service system, promptly. In addition, the administrative and supervisory departments at all levels should transfer the information on the qualifications, credit rewards and punishment, project approval, and punishment for violation of laws and regulations of the parties, to the E-service system within seven working days from the date when the administrative decision is made.

#### 3.1.2. The Vaccine Circulation under Public Resource Trading Platform

Under the “single invoice” system, all procurements of Class-II vaccines, in principle, should be incorporated into a public resource trading platform, which means the county institutions shall purchase vaccines from production enterprises by bidding on the E-trading system. Therefore, all vaccine-related trading information generated in the E-trading system would be transferred to the E-service system, to provide easy public access to the information on vaccine procurements and prevent the recurrence of the Shandong vaccine sales incident.

Besides online procurement through an E-trading system, all provincial institutions are in charge of summarizing the demands for class-II vaccines in their respective administrative regions, and forming reasonable procurement prices on the platform through a direct listing, bidding, or negotiated bargaining. One of the causes related to the vaccine incident in Suqian is the markup charged by either provincial or municipal institutions. Recourse to the bargaining power of provincial institutions, the demand-gathering practice combined with online procurement could fundamentally eradicate overpricing issues of vaccines.

Nonetheless, such an abrupt change of circulation channels caused a massive supply shortage across the country, in that there were still many provinces where the provincial public resources trading platforms had not yet been fully built, and most vaccine production enterprises used to rely on wholesalers for the distribution, barely possessing self-sustained logistic facilities. It has been reported that in the middle and end of June 2016, there was a shortage of rabies vaccine, chickenpox vaccine, influenza vaccine, and other class II vaccines in Hebei, Jiangsu, Henan, Guangdong and other provinces at the same time [25]. Worse yet, in April 2017, Hefei, the capital of Anhui province, experienced a shortage of the five-dose vaccine used to vaccinate against diseases caused by polio, pertussis, diphtheria, tetanus, and Haemophilus influenzae type B [26].

The vaccine supply shortage indicated that the vaccine circulation scheme under the public resource trading platform needed to be further improved with supporting policies. Besides the 8-month interim period for the implementation of the “single invoice” system, State Council suggested that vaccine production enterprises may adopt the segmented relay approach of “trunk transport–regional warehousing–regional distribution” to distribute vaccines. Put another way, the relay approach geographically decomposes the distribution pressure into two parts: production site transportation and area distribution.

### 3.2. The Development of Vaccine Lot Release

The lot release regime in China was initially established in 2004. However, the incident of Changsheng caused public distrust and misunderstanding of the testing rate for vaccine efficacy. In response, Junzhi Wang, Chief Scientist of the China Institute of Biological Products Testing and Certification claimed that vaccines would expire if all had to go through efficacy testing [27]. But how to make institutional adjustments to the testing rate of vaccine efficacy still remained a critical and urgent issue.

In 2017, the China Food and Drug Administration (hereinafter, CFDA) overhauled the previous regulation on the lot release regime and adopted Measures for the Administration of Lot Release of Biological Products (2017) (hereinafter, Measures in 2017), which mainly aims at strengthening the management of lot release agencies, refining the application procedures, highlighting the responsibility of lot release applicants, and enhancing the transparency of the lot release process. As to the efficacy inconsistency between samples for lot release and the stock ones, Article 14 of Measures in 2017 requires the provincial FDA to determine relatively fixed sampling institutions and personnel, make records at lot release institutions, regularly train sampling institutions and personnel, and supervise and guide sampling work. Thus, the fixed sampling institutions and personnel with regular professional training could fundamentally stop applicants from manipulating the sampling procedure.

Concerning the sampling rate of efficacy testing, Guidelines for independent lot release of vaccines by regulatory authorities (hereinafter, Guidelines) were adopted by the 61st meeting of the WHO Expert Committee on Biological Standardization in 2010, which specifies the criteria for the selection of tests for lot release and the percentage of lots to be tested [28]. Specifically, the Guidelines point out that:

If less than 100% of lots are tested, the choice of lots to be tested should be in the hands of the NCL, and the manufacturer should not be aware in advance of which lots will undergo testing.

The percentage of lots tested should be monitored and revised, if necessary, based on experience with the product and data from the yearly biological product report (e.g., good consistency over a significant period may lead to a reduction of the percentage of lots covered, while observance of an undue number of failing results and/or specific testing issues may increase the percentage of lots to be tested).

Given the requirement for the testing percentage by Guidelines, the CFDA made a corresponding amendment to Measures in 2017, of which, Article 18 prescribes that, in the course of the lot release, the institution may conduct a comprehensive evaluation of the technology and quality control, the historical lot release, and other information, to dynamically adjust the inspection items and frequency; if some vaccine product subject to the lot release fails an item, the institution may increase the frequency for subsequent product lots.

Considering the risk of new vaccines, the Guidelines suggests:

Specific attention should be paid to new vaccines (as well as to new manufacturers) for which there is little accumulated experience, and to sophisticated combined vaccines, for which testing and interpretation of results may be complicated.

Similarly, subparagraph (1) of Article 21 of Measures in 2017 stipulates that a whole-item inspection of a product should be conducted according to the registration standards, and a part-item inspection may be conducted only if at least three successive product lots are released, provided the marketing of the product of the lot release applicant is newly approved.

## 4. New Governance in Vaccine Administration

### 4.1. Decentralized or Not: That Is a Question for Vaccine Administration

Prior to the unveiling of healthcare reforms in the 1990s, China had implemented traditional command-and-control regulation for vaccine circulation for several decades. To be more specific, it was the planned mode of vaccine circulation, which required all types of vaccines be purchased uniformly by the Provincial Disease Control and Prevention Institution and supplied level by level based on “Province-City-County” management model, that dominated the vaccine administration in China.

However, since the command-and-control model has long been criticized for its inefficiencies, accompanied by a growing pressure to liberalize the market, new governance theory is brought forward to rejuvenate legal strategies, with the potential to promote the effectiveness and legitimacy of social regulation [29]. The traditional regulatory approach is destined to be ineffective or disruptive in social areas. To begin with, the application of substantive law is likely to be both underinclusive and ineffective in generating significant changes in behavior without running the danger of destroying other subsystems [30]. In addition, the internal structure of social subsystems may also be destroyed by excessive legislation of society [31]. Such a dilemma could be circumvented by using legal strategies that restructure social subsystems rather than only impose substantive orders. As the successor of the traditional regulatory model, new governance addresses the shifting goals and capacities of legal regulation while avoiding the central deficiencies of substantive law. By transforming legal control into a dynamic, reflexive, and flexible regime, the guiding principles of new governance promote other social domains it interacts with to be able to regulate themselves internally [32]. Just as empirical evidence indicates most countries progress from a legal strategy of formal market-based law, to substantive regulatory law, to a governance approach by constructing cooperative relationships with the private market [33].

Compared to the centralized approach in the traditional regulatory mode [34], new governance advocates moving responsibilities outward and downward, to the states, localities, and the private sector [35]. As one of the new governance approaches [29], decentralization indicates, comparing to the centralized control, the responsibilities transferring to localities and private units, just reflecting Justice Brandeis’ theory of laboratory of experimentation [36]. Besides, decentralization also refers to deregulation, i.e., reduction or elimination of regulatory controls, where privatization usually looms out of devolution of regulatory control from the public to the private sector [37]. On the one hand, privatization could be either demand driven, or policy driven. Put it simply, the demand driven privatization means substitute role of private markets for certain public services, while the policy driven one can cause “load shedding” effect, just like the retreatment of government and the radical marketization in China’s healthcare reform in 1990s; on the other hand, in the process of decentralization and privatization, varying amounts of regulatory accountability are transferred to private regulation [38]. Therefore, the effects of privatization will differ depending on how it is implemented and to what extent.

Moreover, decentralization stresses the concept of subsidiarity, which extends to the idea of localness, partiality of knowledge and the challenge of conversion between localities. According to the subsidiarity principle, administrative functions should be performed at the level that is closest to those they will affect [39]. That is, the maximum amount of latitude should be provided by central authority to local discretion to complete the specifics of broadly specified policy. The finest knowledge leading to a potential solution is held by those who are closest to the issue. For this reason, local expertise is required for the precise development and execution of common standards in order to achieve the desired results. Therefore, it is believed that local entities are better positioned than a dominant central entity to oversee activities that impact them.

In essence, decentralization serves as the foundation of public participation, diversity, and competition. As Hayek noted the emergence of division of knowledge, it is inevitable that central authorities delegate as much as possible the discretion to localities, so as to fill the gap of broadly defined policies. In other words, the idea of subsidiarity requires that every governmental task be best carried out at the level that is closest to the people it will affect, and local entities are the ones more suitable to manage functions than the centralized counterpart.

Confronting with the radical healthcare reform in 1990s, the vaccine circulation in China experienced the transformation from the planned mode to decentralized mode, in the context of fully retreating of the state and introduction of the market force. Nevertheless, simple devolution in the field of vaccine administration usually foreshadows severe regulatory disorder, which could be corroborated by China’s vaccine incidents in the past decades. Decentralization is anything but isolation, where multiple links could be generated among subsidiaries, with enhanced engagement contributing to the formation of deliberative and collaborative capacities [40]. In contrast, as discussed in the selected vaccine incidents of China above, China unfroze the control over the access to the vaccine industry in 1990s, which effectuated the license for private and even foreign capitals stepping into the vaccine field, and both vaccine production enterprises and wholesale enterprises were allowed to sell non-immunity-planning vaccines since 2005. But such an arbitrary decentralization of vaccine circulation directly led to the emergence of the jungle of vaccine wholesalers, where vaccine scalping has widely spread across China. For example, both the vaccine incident in Suqian City and Shandong illegal vaccine sales incident are caused by the unqualified wholesalers and slack supervision.

Decentralization and privatization do generate profound effects on the system of social protection in both symbolic and material terms [41]. At the microlevel, the reallocated power and authority to lower levels (including lower units of government and private sector) would inevitably cause tectonic consequences for consumers, such as the unbounded decentralization of vaccine circulation and concomitant vaccine incidents in China. Comparatively at the macrolevel, the decentralization and privatization can be regarded as a conservative endeavor to reorganize society’s primary institutional spheres and to curtail governmental involvement in favor of economic efficiency and individual freedom [42], which cast doubt on the competence of collective action. From the perspective of liberals, they raised a question on centralized bureaucratic institution in social welfare, like the vaccine administration, and call for decentralized local control [43]. Both of conservatives and liberals stress the empowerment of stakeholders in the social system. Nevertheless, it would be a grave mistake to neglecting the implication of progressive empowerment in the reform of social institutions.

### 4.2. Structurally Integrated Decentralization

Just as Tony Judt puts it, “when now we speak of economic ‘reform’ or the need to render social services more ‘efficient’, we mean that the state’s part in the affair should be reduced. The privatization of public services or publicly owned businesses is now regarded as self-evidently a good thing” [44]. However, it is not always self-evident that the state is detrimental for us. Such narrowly construed economism should not be only collectively sought for social or political ends. Judt further pointed out that it was the very success of the mixed-economy welfare states, in providing social stability in the past half century, that has led to the same stability being taken for granted [44].

Undoubtedly the decentralized approach of vaccine administration has managed to address the supply shortage of non-immunity-planning vaccines, and each province and locality has contributed to the evolution of vaccine administration by creating a slew of programs which enact and test reforms, and then subsequently accept or reject them. As a result of enhanced diversity and competition, decentralization and privatization further promote choice and responsiveness for the development of vaccine circulation.

However, the recurring vaccine incidents in the past decades indicate that solely following a decentralized approach to vaccine circulation might not work well in China. In this work, we intend to propose the structurally integrated decentralization model based on the practice of the Public Resource Trading system. In particular, the structurally integrated decentralization is comprised of semi-centralization and semi-decentralization for the vaccine administration, of which semi-decentralization denotes that each province constructs independent public resource trading platforms, and summarize the demands for class-II vaccines in the respective administrative regions; while semi-centralization stresses that each provincial disease prevention and control institution centrally organizes all procurements of non-immunity-planning (or Class-II) vaccines on provincial public resource trading platforms.

In a traditional regulatory environment, the law is often segmented into distinct, specified subfields, by means of a solely localized approach focusing on specified issues in a constricted area. Though it is not surprising that decentralization allows local knowledge to match solutions to contingent situations, decentralization must be structurally integrated with regional and national commitments to orchestrate local efforts in a comprehensive manner. Accordingly, a structurally integrated decentralization model adopts a holistic approach to address problems in a broader context, striving for a comprehensive view on circumstances as they exist simultaneously over a broad disciplinary spectrum [45].

The architecture of structurally integrated decentralization, in essence, emphasizes a certain degree of legal restraint, where instead of taking over regulatory responsibility for the social processes, the law confines itself to the installation, correction, and catalysis of democratic mechanisms [46]. Put another way, the new governance theory regards the legal system as the interaction of the institutions rather than a defined set of principles and rules. Compared to the conventional belief of placing the law at the pulsating core of how social values are expressed, the governance model notices that there is a plethora of social subsystems with numerous cores that occasionally try to interact and coordinate with one another, and the centralized law does not have exclusive authority over all other subsystems. Rather, the law coexists with numerous subsystems while continuously evaluating the viability of the various organizations. Just as the European Autopoiesis school emphasizes this function of the law in establishing the capacities of the various social subsystems, it claims that “legal norms should produce a harmonious fit between institutional structures and social structures rather than influence the social structures themselves.”

In the context of vaccine administration, structurally integrated decentralization even takes a step further by combining semi-centralization and semi-decentralization hierarchically. From the perspective of new governance theory, such a combination, somehow, breaks through the coordinating function of law. Semi-centralization demands that all procurements of vaccines be centrally organized on public resource trading platforms. Such a re-centralization arrangement at the subsystem level is, to some extent, at odds with the coordination approach of new governance theory. By coordinating diverse scales of action, the governance model assists in integrating isolated efforts at the subsystem level, and the coordinating function of law is achieved through its “competence competency”, i.e., the competence to determine other actors’ competencies [7]. On the other hand, structurally integrated decentralization takes advantage of the public resource trading system to compulsively promote broad substantive values across non-legal fields, rather than deferring as much as possible to those subsystems [47]. The public resource trading system is an embodiment of the concept of unified public resource trading, with a gamut of public resources, ranging from bidding for engineering construction projects, granting of land use rights and mining rights, trading of state-owned property rights, and government procurements. All kinds of public resources trading (including Class-II vaccine procurements) on the platform shall conform to the Interim Measures for the Administration of Public Resources Trading Platforms issued by 14 ministries and commissions of the State Council, which elaborately provides rules about the operations, services, supervision and management of the public resource trading platform.

In this framework, structurally integrated decentralization creates a strategic link between local authorities and national objectives, by ensuring that subsystems are responsive to their constituents, and coordinating subsystem operations with the public resource trading system, while maintaining broad subsystem independence, i.e., constructing independent public resource trading platforms and summarizing the demands for class-II vaccines in the respective administrative regions. In this way, the structurally integrated governance strategy establishes a middle ground between actual regulation and deregulation. The law continues to be the critical role in the governance model, but in a different way than it was in the regulatory paradigm, which viewed the law as top-down and universal.

## 5. The Anatomy of Vaccine Administration Law in China

After the aforementioned major vaccine incidents, the Standing Committee of the Thirteenth National People’s Congress adopted the Vaccine Administration Law, which came into force in 2019. Holistically, as shown in Figure 3, the Vaccine Administration Law consists of 8 components throughout the whole life cycle of vaccines: General Provision, Vaccine Development and Registration, Vaccine Production and Lot Release, Vaccine Circulation, Vaccination, Monitoring and Handling of Abnormal Reactions, Management after Marketing of Vaccines and Safeguard Measures and Supervision.

This section mainly focuses on the analysis of vaccine circulation and lot release and attempts to probe into the relationship among internal components of the Vaccine Administration Law. First, Article 10 in General Provision specifies the implementation of a whole-process traceability system for vaccines, including the uniform vaccine traceability standards and the national electronic platform for vaccine traceability, which integrates various information on vaccine production, circulation, and vaccination. As early as 2006, however, the SFDA started to build an electronic drug supervision system for monitoring drug production, circulation, and use [48]. Since 2008, vaccine products have also been incorporated into the electronic supervision system with a 20-digit Electronic Drug Monitoring Code. Due to an abrupt announcement from the SFDA in 2016, the Electronic Drug Monitoring Code was suspended indefinitely. In essence, there were two main causes leading to this outcome: one is the incompatible electronic systems used by supervisory departments; the other is the failure to perform scanning of the Electronic Drug Monitoring Code in most drug wholesalers and Centers for Disease Control and Prevention.

With the implementation of the “single invoice” system for vaccine procurement, and the construction of public resource trading platforms, all vaccine transactions have been converted into a unified electronic system at the provincial level, which solved incompatible system issues for supervisory departments. Furthermore, as part of the construction of a national electronic traceability system, vaccine marketing license holders are required to link self-maintained systems to the national one, realizing traceable and verifiable vaccines of the minimum packaging unit. Equally important, as lessons learned from the failure of the Electronic Drug Monitoring Code, both disease prevention and control institutions and inoculation entities should record vaccine circulation and vaccination, and transfer traceability information back to the national platform.

Based on the whole-process traceability system, the vaccine marketing license holder should accurately and completely record the vaccine transactions, while disease prevention and control institutions and inoculation entities should check the temperature monitoring records of the entire transportation and storage process at the time of receiving the vaccines [49]. During the time of storage, disease prevention and control institutions and inoculation entities should perform periodical inspection in case of packaging damage, non-compliance with temperature requirements, or expiration issues.

Regarding lot release, Article 29 emphasizes the examination of materials and inspection of samples should be conducted on a lot-by-lot basis, distinguishing vaccines from other drug products, while the specific inspection items and frequency could be dynamically adjusted according to the assessment of vaccine quality risks. Aside from the sample inspection, deviation management is proposed as part of the Production Quality Management System for vaccine license holders, which requires enterprises to faithfully record data generated during the processes of production and inspection, including production technology deviations, quality differences, and failures and incidents in the production process, to integrate the data into the documents on the application for lot release. Should there be any doubt over the authenticity of samples, the lot release institution may conduct an on-site sampling inspection for further verification.

As the extension of Production Quality Management, Whole Life-cycle Quality Management also covers the monitoring of abnormal reactions and the management after the marketing of vaccines. For the monitoring of abnormal reactions, vaccine marketing license holders are responsible for collecting, tracking, and analyzing suspected abnormal reactions to vaccination, taking risk control measures on time, and reporting to the disease prevention and control institution. Regarding the management after the marketing of vaccines, vaccine marketing license holders are required to set up vaccine quality retrospective analysis and risk reporting rules, reporting vaccine production and circulation, industry research after marketing, and risk management to the National Medical Products Administration.

## 6. Conclusions

This work has provided a deeper insight into the deficiencies of vaccine circulation and lot release, which prove to be the most critical causes of vaccine incidents. Vaccine circulation has undergone a series of transformations from the unified model to the public resource trading model, where the shift was designed to address the supply-demand imbalance brought about by changes in the vaccine market, while the public resource trading model aims at strengthening the supervision of vaccine distribution channels through the construction of standardized infrastructure platforms. In a nutshell, it is the combination of the stagnant legal system and the absence of information technology infrastructure adaptable to the volume of vaccine production in the process of vaccine administration reform that has led to the recurrence of vaccine incidents.

Furthermore, enacting the Vaccine Administration Law reflects China’s determination to use the strictest management system to safeguard public health and comprehensively improve the drug and vaccine regulatory system. The law systematically outlines a supervision framework throughout the whole life cycle of vaccines, which takes advantage of the Whole Process Electronic Traceability System and Whole Life-cycle Quality Management System to achieve the interconnection of all aspects of the whole life cycle of vaccines.

Revisiting China’s endeavor in vaccine administration so far, there is much room for improvement in vaccine management in the long-term. The reform in vaccine administration is essentially a balance between efficiency and safety, which also represents the interaction between marketization and administrative supervision. The public resource trading model for vaccine circulation means that society pays more. In contrast, some provincial governments in China have already integrated most class-II vaccines into class-I vaccines, instead of strengthening the supervision by constructing public resource trading platforms. So, it is not clear which approach might be more efficient and effective for vaccine administration.

In terms of the participation in global healthcare, WHO Director-General, Margaret Chan, announced the strong results of WHO’s evaluation of China’s National Regulatory Authority for vaccines in 2014, which means that Chinese produced vaccines are quality assured for international standards of production, safety, and effectiveness, a requirement for the procurement of vaccines by UNICEF [50]. In a nutshell, reforms in China’s vaccine administration, combined with the recognition from WHO, represent China’s enduring and active commitment to global public healthcare.

## Figures and Tables

**Figure 1 ijerph-20-03450-f001:**
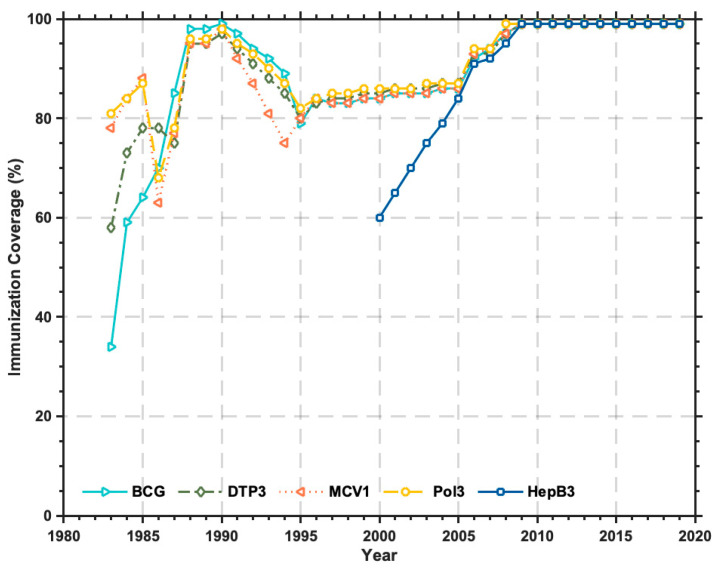
China’s immunization coverage since 1983.

**Figure 2 ijerph-20-03450-f002:**
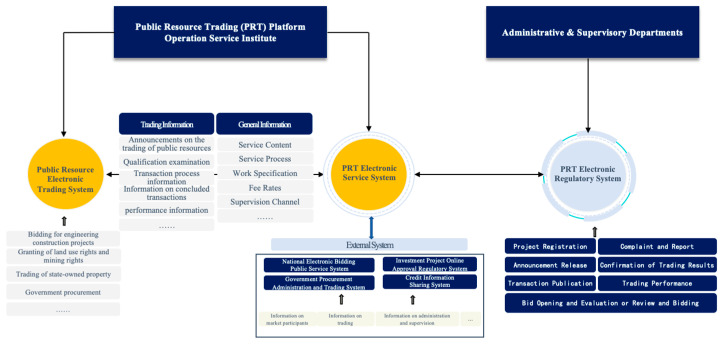
China’s Public Resource Trading Platform System. Source: Interim Measures for the Administration of Public Resources Trading Platforms, http://www.chinatax.gov.cn/chinatax/n810341/n810765/n1990035/201606/c2304070/5118371/files/40d618f83f494486b2586d0405deb7cd.pdf (accessed on 5 February 2023).

**Figure 3 ijerph-20-03450-f003:**
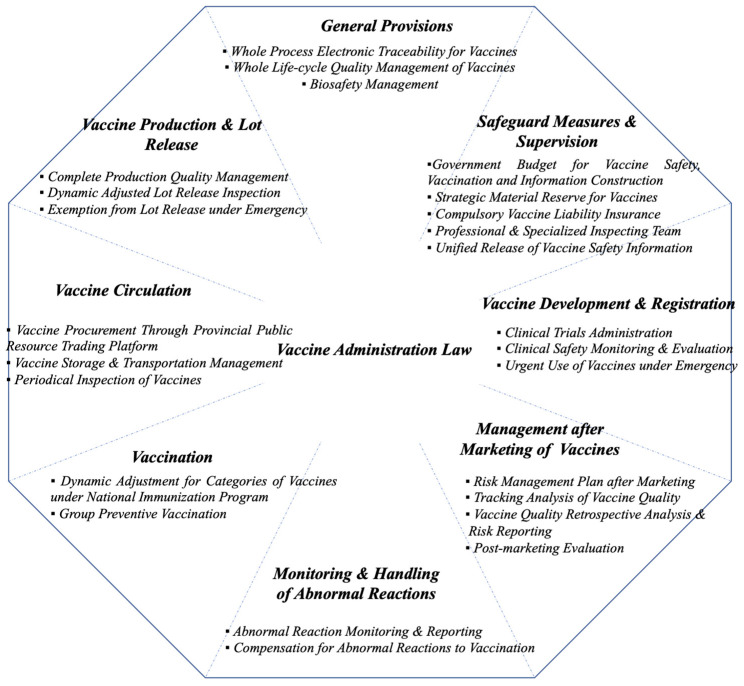
Organization of the *Vaccine Administration Law*.

**Table 1 ijerph-20-03450-t001:** Vaccine incidents in the last two decades.

Time	Location	Vaccine Incident Type
June 2004	Suqian city, Jiangsu Province	Circulation
June 2005	Si county, Anhui Province	Circulation
2007–2010	Shanxi Province	Circulation
February 2009	Dalian City, Liaoning Province	Production & Lot Release
December 2009	Jiangsu Province	Production & Lot Release
September 2012	Weifang city, Shandong Province	Circulation
March 2016	Shandong Province	Circulation
July 2018	Jilin Province	Production & Lot Release

Source: China’s Vaccine Incidents (in Chinese), https://zh.wikipedia.org/w/index.php?title=%E4%B8%AD%E5%9B%BD%E5%A4%A7%E9%99%86%E7%96%AB%E8%8B%97%E4%B9%B1%E8%B1%A1&oldid=73233345 (accessed on 20 June 2022).

## Data Availability

Not applicable.

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
