# Peer review of "Reforms in China’s Vaccine Administration—From the Perspective of New Governance Approach"

_ijerph, 2023, doi:10.3390/ijerph20043450_

Round 1

Reviewer 1 Report

How to improve vaccine administration has been an important issue in China as China has such a huge demand for vaccines. Therefore, the authors raised an interesting question from a perspective of New Governance approach. However, the concept of New Governance should be further clarified. It will be better to use Covid-19 vaccines as a case study to evaluate China's reform of the vaccine administration. Given that China is among the top vaccine manusfacturers in the world, it is worthwhile to examine the implications of China's reform of vaccine administration to the accessibility and affordablity of vaccines worldwide. The lauguage also needs to be improved.  

Reviewer 2 Report

Comments by Reviewer 2

This work reviewed the vaccine incidents happened in mainland China in the past few decades. From an administrative perspective, the authors discussed in detail the multilayered issues and risks hidden or have been exposed in the past. Given the circumstances of the traumatic impacts of COVID-19 on the economy and public health, which is expected to be discussed here, the advice were never more important to China's administrative officials.

Overall, the data were well presented and the authors' arguments were supported by historic statistics.

This paper discussed the multilayered administrative issues that underline the vaccine incidents took place in China over the past few decades, and attempted to come up with resolutions to these issues through administrative reform;

The topic has been discussed before but this paper summarized most of the incidents in detail. This work did cover a broader dimension and provided insightful statistical evidence. Overall, this work could be helpful to the domestic public health officials, whether or not it would lead to any fruitful changes.

Specific improvements the authors should consider:

a)   Source for Table 1. Even though the list is likely correct and reliable, the source is from Wikipedia, which is generally considered a helpful source but not necessarily reliable due to its open access to editing, lack of curation and peer reviews;

b) Throughout the text, a large part of claims were without sources/citations;

Conclusions:

I look forward to seeing the conclusions to be more extensive and be more critical, especially given the multiple recurring incidents throughout the years.

Citations:

Citations need some upgrading to improve the accuracy and credibility of the text.

Figures:

Figure 2. is interesting and new to me, the authors should probably provide the source of this chart, and red-mark the structural elements that needs attention or improvements as discussed in the text.

Reviewer 3 Report

1.       The research topic is of great significance. However, the author does not seem to provide any new research conclusions. 2.The explanation of the new governance in the article is not clear, and it is suggested to make more specific explanation.3. Vaccine management is not only the balance between safety and efficiency, but also needs to be paid attention to and studied from a more comprehensive perspective. For example, the development of medical science itself provides choices for the management. 4. The author lacks in-depth understanding of the regulatory science abroad, It is suggested that the literature review should be supplemented. 5. The author's research methods are not enough to support the research issues. The author's case selection in the article is also relatively simple, and there is also a lack of in-depth field work on the specific vaccine approval and regulatory system of the State Pharmaceutical Administration of China. It is suggested that the work can be do more in-depth.

Reviewer 4 Report

Some revision suggestions as follows.

It is suggested to highlight the importance of timely information disclosure about the whole process from vaccine manufacture, procurement, to distribution. 

I am not quite sure about the meaning of the sentence in lines 37, 38. Especially the accounts. Could you please briefly explain?  

Could you please explain the meaning of the sentence in lines 52, 53?

Suggest making the correction at line 57, a systemic view of... into a systemic view on. 

If possible, could you please clearly explain if the children from destitute areas achieved the complete vaccination procedure of the HBV vaccine, and all types of vaccines that were suggested by the Expanded Program at that time? 

Suggest making a revision at lines 98, 99. ... who provided the invoice from Chuzhou Pharmaceutical Technology Development Corporation rather than the official channels from Anti-Epidemics Station. This corporation was just run as a drug wholesaler...... 

Suggest making a revision at line 106. There are two main causes, ...

Please explain 'the vaccine monopoly in the 1990s' on line 107. CBPGC was established in 1989, as mentioned in line 136.

Please confirm the expression of the sentence in lines 130, and 131. After knocking down the budget, the payroll accounted for 50% of the total government subsidy received by the center in 2004. I can't get it. Please don't make any changes if it were right.

Please confirm the use of 'the authenticity of samples' here in line 163. As mentioned above, the sampling procedure completed by accredited staff from the authorized third party is decisive. In addition, I am curious, could you explain the use of 'used to be' here? 

The use of coincidently in line 225 reads a little bit intriguing. Is another word possibly more accurate?

Suggest making a revision at line 243, ......to vaccinees, thereafter even vaccination hesitancy..... 

Please confirm that the abbreviations PRT or PRTP, FDA et al., were defined at their first occurrence.

Please reorganize Figures 2 and 3 with font larger in the same size. 

Suggest deleting 'some' at line 376. 

Suggest changing 'view of circumstances' at line 481 into 'view on circumstances'. 

'....that decentralization allows local .......' at line 477.

Could you please further explain the sentence in lines 489, and 490? 

Some sentences in those paragraphs from lines 497 to 526 are quite long and suggested appropriately adding the full stop in between. 

In line 598, it is easier to understand the state's determination to strictly manage the vaccines. Maybe it is still hard for us to imagine the strictest. 

Suggest making a revision in line 608, '......vaccine circulation means that the society pays higher ......'.

In line 614, I am interested in the strong results of WHO's evaluation work. Could you further explain it in the article. 

Round 2

Reviewer 3 Report

The thesis has improved a lot, and still needs some improvement.General some typing errors and the language should be looked carefully again.You should offer some theoretical structure for these dimensions you have measured.The  presentation of the method  is not  clear as now.The method part can be described more clearly.
